# Differences in the clinical presentation of sleep apnea patients according to age and gender

**Olli Lyyra** [1,2*], **Aino Lammintausta**[2], **Per-Erik Gustafsson**[3,4], **Ulla Anttalainen**[2,4‡],
**Tarja Saaresranta**[2,4‡]

1 Southwest Finland Wellbeing Services County, Raisio Health Center, Raisio, Finland, 2 Sleep Research Centre, University of Turku, Turku, Finland, 3 Turku University Hospital, Auria Clinical Informatics, Turku, Finland, 4 Division of Medicine, Department of Pulmonary Diseases, Turku University Hospital, Turku, Finland

‡ TS and UA share the last authorship.
* ollyyr@utu.fi

## Abstract

### Background

Data on sleep apnea is scarce in the elderly. We aimed to provide insight into the presentation of sleep apnea in people over 70 years of age including gender differences.

### Methods

We conducted a registry study in sleep apnea patients >18 years of age diagnosed at Turku University Hospital in 2012–2019. Patients whose sleep apnea was classified at least moderate according to apnea-hypopnea index were included (N = 5870; Men = 65.7%; Mean age 57.5 ± 13.5 years). Data on cardiorespiratory polygraphy (PG) variables, Body Mass Index (BMI), pre-existing depression diagnoses, number of comorbidities, the scores of Epworth Sleepiness Scale (ESS), depression scale (DEPS) and psychological distress (12-item General Health Questionnaire, GHQ-12), and capillary blood gas results were derived from electronic medical records. Patients were stratified into three groups according to age: <70 (young-middle aged), 70–80 (elderly) and >80 years of age (very elderly).

### Results

The severity of sleep apnea did not differ between the age groups based on any of the PG variables studied. No significant differences were found in the level of subjective daytime sleepiness between age groups. Women had higher DEPS scores than men in all age groups. Very elderly men had higher DEPS scores compared to men in other age groups (6.3 ±4.6 vs. 5.6 ±5.9 vs. 5.1 ±4.8, p < 0.05) while the differences in DEPS scores did not reach significance among women. Each unit increase in $SpO_2$ was associated with a 22% decrease in the odds of having a DEPS score of ≥9.

**Data availability statement:** All relevant data are within the paper and its Supporting Information files.

**Funding:** OL, AL, UA, TS: Väinö and Laina Kivi Foundation (https://www.foundationweb.net/kivi/), Turku University Hospital Governmental VTR Grant 13542 (https://www.utu.fi/fi/tutkimus/tutkimusrahoituksen-tuki), Finnish Anti-Tuberculosis Foundation (https://www.tb-foundation.org/), Tampere Tuberculosis Foundation (https://www.tuberkuloosisaatio.fi/), The Research Foundation of the Pulmonary Diseases (HES Foundation) (https://www.hes-saatio.fi/). The funding sources had no role in the design and conduct of the study; collection, management, analysis, and interpretation of the data; preparation, review, or approval of the manuscript; and decision to submit the manuscript for publication.

**Competing interests:** The authors have declared that no competing interests exist.

## Conclusion

The severity of sleep apnea or subjective daytime sleepiness did not differ among age groups in moderate-severe sleep apnea patients. Occurrence of depressive symptoms was consistently more common in women than in men of comparable age. Mental well-being was the worst in the very elderly. Higher $SpO_2$ was associated with less depressive symptoms.

## Introduction

Sleep apnea is an increasingly prevalent public health problem due to aging and the obesity epidemic. According to the United Nations' estimates, 143 million people are today aged 80 years or older. By 2050, there will be nearly 430 million 80-year-olds worldwide [1].

Sleep apnea in the elderly is associated with several adverse health consequences and a wide variety of comorbid conditions [2–4]. Elderly people partially share the same sleep apnea symptom profile as middle-aged individuals [4]. However, it is possible that cognitive impairment, nocturnal urination and recurrent falls can be the main complaints in elderly individuals [2–6]. According to previous studies the cardinal symptom of sleep apnea, excessive daytime sleepiness (EDS), is often absent or at least experienced milder in the elderly [7–9]. The symptoms of sleep apnea can also be confused with the physiological changes related to aging, such as a decline in sleep quality and cognition.

Depression is another important public health concern that is associated with a great deal of human suffering and high healthcare costs. Multiple studies have linked obstructive sleep apnea with depression and a range of other psychiatric disorders [10–14]. The prevalence of depressive symptoms and clinically diagnosed depression are higher among patients with sleep apnea than in those without it [15]. There is some evidence of comorbidity between depression and sleep apnea also in the elderly [16,17] However, deeper understanding of the direction of association between these conditions and their impact on elderly patients' health is needed.

The commitment of the elderly to continuous positive airway pressure (CPAP) treatment has long been of interest. Concomitant depressive symptoms may however impair adherence to CPAP treatment [18]. In population studies, the prevalence of major depression ranges from 0.9% to 42% among elderly [19] whereas prevalence of sleep apnea in older people in the general population can vary from 20% all the way to 80% [20–23]. Finnish population studies have estimated 12-month prevalence of clinically significant depression in the adult population and young people to be approximately 5–7% and slightly lower in the over 65-year-olds [24,25]. In a study by Markkula et al. the 12-month prevalence of depressive disorder was as high as 8% in those aged 75 or older [25]. In a study published by Acker et al. the prevalence of depression was found to be 21.5% among adult people with sleep apnea [26,27]. Given the high prevalence rates and shared symptoms of these conditions we may end up in misdiagnosing sleep apnea for depression or vice versa.

There are also limited data regarding gender differences in symptoms, comorbidities as well as cardiorespiratory polygraphy (PG) variables of elderly sleep apnea patients. It has been stated that women are more likely to suffer from excessive daytime sleepiness and depressive symptoms [28–30]. Whether this is also true in elderly patients is unknown. Multimorbidity in individuals with sleep apnea is highly prevalent [27] but data on the burden of comorbidities especially in the elderly sleep apnea patients is scarce. We wanted to find out how multimorbidity, age and gender affect the symptoms of sleep apnea in the elderly and enhance especially the understanding of role of depression in the elderly sleep apnea patients.

## Methods

### Patients and data extraction

We conducted a comprehensive retrospective registry study of consecutive patients aged 18 years and over diagnosed with sleep apnea at Turku University Hospital in years 2012–2019. Patients with sleep apnea were identified from electronic medical records using G47.3 ICD-10 code (International Statistical Classification of Diseases and Related Health Problems, 10th edition). Out of the 13,651 patients found, we included patients for whom sleep registration data could be found at least partially and whose sleep apnea based on the apnea–hypopnea index (AHI) was at least moderate (AHI >15).

The study was approved by the Turku University Hospital Clinical Research Center which is our Institutional Review Board (Approval number: TO5/040/19). Patient integrity was not compromised as we used anonymised patient hospital records without contacting the patients, and therefore the permission of the ethics committee or the patient's informed consent was not required according to Finnish legislation. Auria Clinical Informatics (Turku University Hospital, Turku, Finland) collected data from electronic medical records between April 21, 2020 and December 5, 2022. The data was accessed for research purposes for the first time on August 8, 2020.

In accordance with local practice, the majority of the PG examinations are performed in primary care or occupational health care. Thereafter, patients diagnosed with sleep apnea are referred to specialist care if lifestyle modifications and positional treatment are judged to be inadequate. A smaller proportion of PG's are performed at the Department of Pulmonary diseases or Clinical Neurophysiology, most of them as ambulatory investigations. Less than one percentage of patients are diagnosed with polysomnography (PSG).

PG or PSG data from patients diagnosed with sleep apnea as well as anthropometrics, data from capillary blood gas analysis, ESS score [31], depression scale score (DEPS) [32], and the 12-item General Health Questionnaire score (GHQ-12) [33] were collected from electronic medical records using a search algorithm. Values for all variables were obtained, if possible, from the start date of sleep apnea treatment, or from the date closest to start date.

Sleep studies were scored according to the rules of the American Academy of Sleep Medicine valid at the time of performed sleep studies. AHI was used to describe sleep apnea severity. Hard copies of sleep study reports from PGs performed elsewhere than the hospital were not included in electronic health records. Furthermore, medical narratives were often short and not including all data of sleep study reports resulting in missing data.

Mean and minimum values of peripheral oxyhemoglobin saturation ($SpO_2$) measured by pulse oximeter were obtained from sleep studies. ESS was used to assess subjective daytime sleepiness. Usually, 10 points or more is interpreted as clearly abnormal, indicating EDS. DEPS [32] is a tool for identifying depressive symptoms and screening for depression. About one-third of those scoring 9 points or more on DEPS, and nearly half of those scoring 12 points or more have depression. GHQ-12 [33] measures psychological distress. If the subject receives 3 points or more, the test is "positive", in which case the sensitivity for the diagnosis of mental illness is about 70% and the specificity is about 75%.

For assessing multimorbidity and heavy multimorbidity, we retrieved comorbidity data from the patient data system. We obtained data on diagnoses that were present prior or at the time of sleep apnea diagnosis. We then classified the diseases relevant to sleep apnea according to the classification introduced by Palomäki et al. [27] which is, in turn, based on models introduced by the Finnish Institute for Health and Welfare and Calderón-Larrañaga et al. [34].

An individual was classified as multimorbid if two or more chronic diseases were present prior or at the time of sleep apnea diagnosis and heavily multimorbid if they had at least four

chronic diseases. Comorbidity data was available from 2004 to 2019. The ICD-10 codes were available with a precision of the first three characters. For this study, we also collected information about the diagnosis of depression with ICD-10 codes F32, F33 and F34.

Capillary blood partial pressure of carbon dioxide (CB-pCO$_2$) is generally determined especially in obese and chronic obstructive pulmonary disease (COPD) patients with sleep apnea to identify concomitant hypercapnic respiratory failure such as obesity hypoventilation syndrome.

## Statistical analysis

For descriptive statistics and basic data analysis, patients were divided into three age groups: <70 (young-middle aged adults), 70–80 (elderly) and those over 80 years of age (very elderly).

Descriptive clinical characteristics of the sample were summarized by listing means and standard deviation for continuous variables, unless otherwise stated. Continuous variables were compared between two age groups using the non-parametric Mann-Whitney U test, and with the Kruskal-Wallis H test for multiple comparisons. The Chi-square test was used to compare categorical variables between age groups. From the multiple comparisons we reported the Bonferroni corrected adjusted p-values.

Univariate and multivariate logistic regression was conducted to analyze the effects of age, gender, AHI, mean SpO$_2$, minimum SpO$_2$, CB-pCO$_2$, BMI, ESS score, pre-existing diagnosis of depression and multimorbidity on the likelihood of having DEPS score at least 9. We reported data as odds ratio (95% confidence interval). Patients at least 70 years of age at the time of diagnosis were included in the regression analyses. Missing values were handled by available-case method in statistical analyses.

The p-value < 0.05 was considered statistically significant. All statistical analyses were performed using the SPSS software (IBM Corp. Released 2020. IBM SPSS Statistics for Macintosh, Version 27.0. Armonk, NY: IBM Corp).

## Results

A total of 5870 patients were included in the study (Men = 3855, 65.7%; mean age 57.5 ± 13.5 years). The age range of the patients was 18–101 years. There were 253 (4.3%) patients over 80 years of age (very elderly), 919 (15.7%) patients ranging in age from 70 to 80 years (elderly) and 4698 (80.0%) were aged under 70 years (young-middle aged) at the time of sleep apnea diagnosis. BMI differed between these two age groups in both men and women (p < 0.05). There was no difference in BMI between the elderly and the very elderly men or women (Table 1).

The severity of sleep apnea did not seem to differ between age groups in terms of AHI or mean and minimum values SpO$_2$. AHI was somewhat higher in men compared to women in every age group (p < 0.05) (Table 1).

Young-middle aged women had higher ESS scores (7.9 ± 4.9 vs. 7.4 ± 4.6, p < 0.05) compared to young-middle aged men. Apart from that, there did not seem to be any differences between the groups in the perceived daytime sleepiness (Table 1).

GHQ-12 scores were higher in the very elderly compared to other age groups (3.4 ± 3.3 vs. 2.9 ± 3.6 vs. 2.6 ± 3.4, p < 0.05) suggesting higher levels of mental distress. The same difference was also observed in S1 Table when a cut-off value of at least 3 points was used for the comparison of GHQ-12 scores.

Using the same cut-off value we found a significant difference between the genders in the young-middle aged and elderly age groups (p < 0.05) but not in the very elderly (Table 1 and S1 Table).

**Table 1. Relationship between gender and clinical characteristics of sleep apnea stratified by age.**

| | | Age group | | | | | | |
|---|---|---|---|---|---|---|---|---|
| | | Young-middle aged <70 (N = 11184) | N | Elderly 70–80 (N = 2010) | N | Very elderly >80 (N = 457) | N | P |
| AHI | All | 38.7 ± 21.5 | 4698 | 35.6 ± 15.8 | 919 | 37.1 ± 17.5 | 253 | .433 |
| | Males | 40.5 ± 21.9* | 3192 | 36.2 ± 15.8 | 522 | 38.2 ± 15.6* | 141 | .045 |
| | Females | 34.9 ± 20.1* | 1506 | 34.9 ± 15.8 | 397 | 35.8 ± 19.6* | 112 | .066 |
| Mean SpO$_2$ | All | 92.1 ± 2.4 | 2380 | 91.8 ± 2.1 | 429 | 91.6 ± 2.8 | 146 | .001 |
| | Males | 92.0 ± 2.5* | 1627 | 91.8 ± 2.2 | 243 | 92.0 ± 2.4 | 82 | .235 |
| | Females | **92.4 ± 2.2*** | 753 | 92.0 ± 2.0 | 186 | 91.2 ± 3.2 | 64 | <.001 |
| Min. SpO$_2$ | All | 78.2 ± 7.6 | 1990 | 78.0 ± 7.2 | 314 | 77.7 ± 7.9 | 63 | .608 |
| | Males | 78.0 ± 7.9 | 1365 | 77.9 ± 6.5 | 175 | 78.9 ± 7.0 | 33 | .572 |
| | Females | 78.7 ± 7.1 | 625 | 78.1 ± 8.1 | 139 | 76.3 ± 8.8 | 30 | .291 |
| BMI | All | **33.8 ± 6.9** | 4123 | 30.91 ± 5.8 | 849 | 30.4 ± 5.7 | 225 | <.001 |
| | Males | **33.1 ± 6.2*** | 2786 | 30.1 ± 5.2* | 484 | 29.1 ± 4.9* | 126 | <.001 |
| | Females | **35.5 ± 7.8*** | 1337 | 31.9 ± 6.3* | 365 | 32.0 ± 6.2* | 99 | <.001 |
| ESS | All | 7.5 ± 4.7 | 3692 | 6.8 ± 4.3 | 694 | 7.1 ± 4.3 | 200 | <.001 |
| | Males | 7.4 ± 4.6* | 2524 | 6.5 ± 4.1 | 398 | 7.2 ± 4.3 | 112 | .005 |
| | Females | 7.9 ± 4.9* | 1168 | 7.1 ± 4.6 | 296 | 7.2 ± 4.5 | 88 | .025 |
| DEPS | All | **6.3 ± 6.3** | 2996 | **6.4 ± 5.5** | 567 | 7.2 ± 5.4 | 185 | <.001 |
| | Males | **5.6 ± 5.9*** | 2038 | **5.1 ± 4.8*** | 316 | 6.3 ± 4.6* | 106 | .012 |
| | Females | 7.8 ± 6.7* | 958 | 8.0 ± 5.9* | 251 | 8.6 ± 6.2* | 79 | .161 |
| GHQ-12 | All | **2.9 ± 3.6** | 3587 | **2.6 ± 3.4** | 673 | 3.4 ± 3.3 | 194 | .003 |
| | Males | 2.6 ± 3.5 | 2459 | **2.3 ± 3.1*** | 385 | 3.0 ± 3.3 | 110 | .035 |
| | Females | 3.4 ± 3.7 | 1128 | 3.1 ± 3.6* | 288 | 3.7 ± 3.3 | 84 | .086 |
| CB-pCO$_2$ | All | 5.3 ± 0.5 | 2898 | 5.2 ± 0.5 | 630 | 5.3 ± 0.7 | 181 | .066 |
| | Males | **5.3 ± 0.5** | 1953 | 5.2 ± 0.5 | 346 | 5.1 ± 0.6* | 97 | <.001 |
| | Females | **5.3 ± 0.5** | 945 | **5.3 ± 0.6** | 284 | 5.5 ± 0.7* | 84 | .003 |

Data are presented as mean ± SD (Standard Deviation). N = valid values for each variable. Bold indicates that there is a significant difference (p < 0.05) between the means of bolded and the oldest age group.

*indicates P < 0.05 between genders. Abbreviations: AHI = Apnea-hypopnea index/h; BMI = body mass index (kg/m2); CB-pCO$_2$ = Capillary blood partial pressure of carbon dioxide (kPa); DEPS = The Depression Scale score; ESS = Epworth sleepiness scale score; GHQ-12 = 12-item General Health Questionnaire score, SpO$_2$ = Peripheral oxyhemoglobin saturation measured by pulse oximeter (%).

The DEPS scores tended to increase with age. Very elderly men had higher DEPS scores compared to other age groups (p < 0.05) while no significant was found in DEPS scores between the age groups in women. DEPS scores were, however, higher in women than in men in every age group (p < 0.05). The difference in DEPS score between the genders remained significant when a cut-off value of at least 9 was used (p < 0.05) (Table 1 and S2 Table).

CB-pCO$_2$ were slightly higher in young-middle aged men compared to very elderly (5.3 ± 0.5 vs. 5.1 ± 0.6, p < 0.05). In contrast, women's capillary blood carbon dioxide values were higher in the oldest age group (5.5 ± 0.7 vs. 5.3 ± 0.6 vs. 5.3 ± 0.5, p < 0.05) (Table 1).

Table 2 shows the proportions of obese sleep apnea patients. Of those under 70 years of age (young-middle aged) 68.2% were obese (mean BMI 33.8 ± 6.9 kg/m²), whereas of the very elderly 48.0% had BMI ≥ 30 (30.4 ± 5.7 kg/m²).

Table 3 demonstrates the proportions of multimorbid and heavily multimorbid patients in the three age groups at time of sleep apnea diagnosis. As could be expected,

**Table 2. The proportion of obese among sleep apnea patients by age group and gender.**

| | Age group | | |
| --- | --- | --- | --- |
| | Young-middle aged <70 | Elderly 70–80 | Very elderly >80 |
| Men | **1836 (65.7%)**[*] | 222 (45.9%)[*] | 50 (39.1%) |
| Women | **985 (73.6%)**[*] | 221 (60.5%)[*] | 59 (59.6%) |

Bolded values indicate a significant difference between the bolded and the oldest age group.

[*]indicates significant difference between genders at each age group.

**Table 3. Proportions of multimorbid and heavily multimorbid patients in the three age groups.**

| | Age group | | | | | | p |
| --- | --- | --- | --- | --- | --- | --- | --- |
| | Young-middle aged <70 | | Elderly 70–80 | | Very elderly >80 | | |
| | N | Percent | N | Percent | N | Percent | |
| Multimorbidity all | 2915 | **62.0%** | 768 | **83.6%** | 229 | 90.5% | <.001 |
| Men | 1844 | **57.8%**[*] | 434 | **83.1%** | 129 | 91.5% | <.001 |
| Women | 1071 | **71.1%**[*] | 334 | 84.1 | 100 | 89.3% | <.001 |
| Heavy multimorbidity all | 1353 | **28.8%** | 520 | **56.6%** | 182 | 71.9% | <.001 |
| Men | 804 | **25.2%**[*] | 308 | **59.0%** | 99 | 70.2% | <.001 |
| Women | 549 | **36.5%**[*] | 212 | **53.4%** | 83 | 74.1% | <.001 |

Bold indicates that there is a significant difference (p < 0.05) between the means of bolded and the oldest age group.

[*]indicates significant difference between genders.

the proportion of multimorbid and heavily multimorbid people increased with age. There was no significant difference in the proportions between genders in the two oldest age groups.

Table 4 shows the prevalence of depression diagnosis prior or at the time of sleep apnea diagnosis according to age group and gender. The very elderly men and women had a higher prevalence of depression compared to the young-middle aged (p < 0.05). The prevalence of depression seemed to decrease linearly in the entire study population with increasing age. However, there was not significant difference between the two oldest age groups. The prevalence of depression was higher in women than in men in the two youngest age groups (p < 0.05).

In the univariate regression model a significant positive interaction was found between a DEPS score of at least 9 and age, female gender, CB-pCO$_2$, ESS score and pre-existing depression diagnosis. In the univariate model the individuals with a pre-existing depression diagnosis had a 2.66 times greater odds of having DEPS score at least 9. Female gender seemed to be the second strongest determinant (OR 2.23, p < 0.0019) (Table 5).

All 10 variables were simultaneously entered in the multivariate model. The multivariate model was statistically significant, $\chi^2(10) = 157.63$, p < 0.001. The multivariate model explained 16.8% (Nagelkerke R$^2$) of the variance in DEPS score at least 9 classifying 72.8% of the cases correctly. In the multivariate model the effect of 2 pre-existing depression diagnosis diminished, and was not a significant determinant (p < 0.650). Females had 3.25 times higher odds of having DEPS score ≥ 9. Mean SpO$_2$ was another significant predictor (p = 0.004), with each unit increase in SpO$_2$ associated with a 22% decrease in the odds of having DEPS score ≥ 9. The ESS score was also significant with each unit increase leading 18% increase in the odds of having a DEPS score ≥ 9 (Table 5).

**Table 4. Prevalence of depression diagnosis prior or at the time of sleep apnea diagnosis according to age group and gender.**

|  | Age group | | |
|---|---|---|---|
|  | **Young-middle aged** **<70** | **Elderly** **70–80** | **Very elderly** **>80** |
| Men | **234 (7.4%)**[*] | 16 (3.1%)[*] | 2 (1.4%) |
| Women | **246 (16.3%)**[*] | 25 (6.3%)[*] | 4 (3.6%) |

Bolded values indicate significant difference between the bolded and the oldest age group.

[*]indicates significant difference between genders at each age group.

**Table 5. Determinants of DEPS score ≥ 9. Univariate and multivariate logistic regression analysis.**

|  | Univariate | | Multivariate | |
|---|---|---|---|---|
|  | **OR (95% CI)** | **p-value** | **OR (95% CI)** | **p-value** |
| Age | 1.05 (1.02–1.09) | **0.001** | 1.02 (0.95–1.11) | 0.555 |
| Gender (Female) | 2.23 (1.63–3.03) | **<.001** | 3.25(1.59–6.63) | **0.001** |
| AHI | 1.01 (0.99–1.02) | 0.224 | 1.00 (0.97–1.02) | 0.638 |
| Mean $SpO_2$ | 0.97 (0.86–1.05) | 0.414 | 0.78 (0.66–0.93) | **0.004** |
| Min. $SpO_2$ | 0.99 (0.96–1.02) | 0.516 | 1.02 (0.97–1.08) | 0.386 |
| CB-$pCO_2$ | 1.43 (1.03–1.97) | **0.032** | 0.85 (0.41–1.75) | 0.649 |
| BMI | 1.02 (0.99–1.05) | 0.210 | 0.98 (0.92–1.04) | 0.447 |
| ESS | 1.13(1.09–1.17) | **<.001** | 1.18 (1.09–1.27) | **<.001** |
| DEPS | 2.66 (1.26–5.61) | **0.011** | 1.41 (0.32–6.12) | 0.650 |
| Multimorbidity | 1.25 (0.81–1.92) | 0.310 | 1.10 (0.44–2.76) | 0.839 |

Data are presented as odds ratio (95% confidence interval). Abbreviations: AHI = Apnea-hypopnea index/h; BMI = body mass index (kg/m2); CB-$pCO_2$ = Capillary blood partial pressure of carbon dioxide (kPa); DEPS = The Depression Scale score; ESS = Epworth sleepiness scale score; $SpO_2$ = Peripheral oxyhemoglobin saturation measured by pulse oximeter (%).

## Discussion

The present study provides novel data of older adults with sleep apnea and also on gender differences. To the best of our knowledge, compared to previous similar studies comparing age and gender differences of sleep apnea, our study included the highest proportion of persons over the age of 80 ("very elderly").

Previous studies have shown that the link between body weight and sleep apnea deteriorates to some extent with age [35,36]. The results of our study suggest that almost half of the very elderly with at least moderate sleep apnea are obese, but in line with previous findings the elderly were less obese than the young-middle-aged.

It is known that the prevalence of sleep apnea tends to rise with age [22]. We do not know to what extent the increased prevalence of sleep apnea is due only to the physiological increase in AHI with aging [37]. It is also poorly known whether there is a ceiling effect with age for this rising tendency of AHI and, how does the increasing severity of sleep apnea affect the symptoms in different age groups and gender. For this study, we selected patients whose AHI was over 15, i.e., the severity of sleep apnea was at least moderate to reduce the effect of physiological AHI increase on our research results. The severity of sleep apnea did not differ between age groups based on any of the PG variables studied. AHI was, however, somewhat higher in men than in women in every age group.

It has been stated that older patients suffer less from daytime sleepiness [8]. However, most studies use ESS as a measure of EDS, which may underestimate the symptoms in older adults

with sleep apnea. In a study by Iannella et al. persons over 65 years of age had a lower average ESS compared with the younger ones [8]. Our study is the first to compare over 80-year-olds to younger age groups. The patients in our cohort were surprisingly homogeneous not only in terms of PG measurements but also regarding daytime sleepiness even when considering the selectivity of the group. Young-middle aged women had higher ESS scores compared to young-middle aged men. Apart from that, there did not seem to be any differences between the groups.

In addition to daytime sleepiness, sleep apnea can manifest as insomnia, impaired quality of life and depression. Depressive symptoms in sleep apnea are more common in women [28,29,38]. There is some evidence that depression is recognized as a comorbidity also in the elderly [16,17]. To our knowledge there is no data on the prevalence of depression diagnosis in elderly sleep apnea patients. Sforza et al. investigated the rate of depressive symptoms from healthy elderly with unrecognized obstructive sleep apnea and found out that neither the existence nor the severity of the sleep apnea was associated with anxiety or depression score [39].

Different questionnaires are widely used in studies investigating depression or depressive symptoms in sleep apnea patients. However, the actual diagnosis of depression cannot be based only on symptom questionnaires. Self-reported symptoms do not reliably differentiate between transient depressive symptoms and DSM-5 criteria of depression. The diagnosis of depression is always based on the symptoms found during the interview, the number, severity, duration and temporal predominance of them, and ruling out differential diagnoses with physical examination and laboratory tests. In our study levels of depressive symptoms were consistently higher in women than men of comparable age including the very elderly, which is in line with the previous studies [28,29,38]. In men, the DEPS score was higher in the very elderly than in other age groups (p < 0.05). For women, the DEPS score also appeared to be slightly higher in the very elderly than in younger age groups. Interestingly, in contrast, there was a decreasing trend in the number of depression diagnoses with increasing age in both men and women. For both men and women, the number of depression diagnoses differed in the youngest age group compared to others. We also examined the prevalence DEPS scores of at least 9 by age group and gender. In very elderly women 46.8% of patients had DEPS score of at least 9, whereas in very elderly men, 32.1% achieved this score. It has been studied that about one-third of those scoring 9 points or more on the DEPS, have depression [32]. We can speculate that among moderate-severe sleep apnea patients about 15% of very elderly women and 10% of very elderly men are likely to have depression. When compared the proportions with pre-diagnosed depression, the gap is significant. Considering that ICD-10 code F34 includes not only long-term depression but also other long-term mood disorders, this difference is even more significant.

It is possible that a larger proportion of elderly depression remains undiagnosed in the elderly, perhaps due to prejudice that depression is a normal part of ageing. Indeed, a large Brazilian National Survey with 70,806 participants (15–107 years old) found that the prevalence of depression underdiagnosis was higher in the among elderly population [40]. On the other hand, as mentioned, the symptoms of depression and sleep apnea can resemble each other, especially in the elderly. Symptom questionnaires may also somewhat overestimate the difficulty of the situation. The patient data for our study was from specialized medical care. It is possible that it underestimates the prevalence of depression, especially milder depression. On the other hand, the situation is the same for all age groups.

GHQ-12 scores were higher in the very elderly compared to other age groups meaning higher levels of mental distress. The same difference was also observed when a cut-off value of at least 3 points was used for the comparison of GHQ-12 scores. Using the same cut-off value we found a significant difference between the genders in the young-middle aged and elderly age groups.

Partial pressures of carbon dioxide from capillary blood were slightly higher in young-middle aged men compared to very elderly which probably is the result of obesity and subsequent hypoventilation being more frequent in the young-middle-aged. In contrast, women's CB-pCO$_2$ values were higher in the oldest age group which can be thought to be impact of late menopause on ventilatory control [41].

As expected, the proportion of multimorbid and heavy-multimorbid people increased with age. There was no significant gender difference in the two oldest age groups.

Lastly, we wanted to find out in more detail which factors influence the appearance of depressive symptoms in elderly and very elderly sleep apnea patients. In the multivariate analysis, the female gender stood out by being the most significant factor influencing the occurrence of depressive symptoms. Interestingly, mean SpO$_2$ also contributed to the occurrence of depressive symptoms, although AHI or multimorbidity had no effect. We also found a link between depressive symptoms and daytime sleepiness. It is not surprising that daytime sleepiness is also reflected in mood. Condoleo et al. showed a significant improvement in depressive symptoms after treatment of sleep apnea with CPAP [42]. However, it is also possible that daytime fatigue is caused by concomitant depression and is not directly related to sleep apnea.

Our study highlights the impact of age and gender on the presentation of sleep apnea in the elderly population. More detailed exploration of different presentation of phenotypes and their impact on treatment adherence as well as treatment efficacy is needed in order to focus the limited healthcare resources cost-effectively.

## Strengths and limitations

There are some major strengths in our study. First, we utilized a large number (N = 5870) of real-world patient data for these analyses, which included a high percentage of females. Second, the proportion of the very elderly (>80-year-olds) in this dataset is, to our knowledge, higher than that of any other similar study. Third, in addition to cardiorespiratory polygraphy measurement and different symptom questionnaires, our study also included information on anthropomorphic measurements such as BMI. We also had comorbidity data at our disposal.

There are, however, also limitations of this study that need to be considered. Firstly, we did not have a healthy control group available, therefore we cannot draw definite conclusions whether the patients' symptoms are due to sleep apnea or some other unknown factor. However, our intention was to compare, sleep apnea symptoms and other variables by age group and gender, and we think we succeeded in that. Secondly, PSG is the gold standard when diagnosing sleep apnea, and more accurately determines the severity of sleep apnea. Therefore, we cannot exclude that using PG may overlook or weaken some differences between the age groups or genders. Thirdly, due to the retrospective study design, there was a high number of missing values in most variables. However, AHI was available for all patients included in the study, so we were able to compare patients by sleep apnea severity and deliberately selected only patients with moderate to severe sleep apnea, ranked by AHI. The patient data of our cohort consists of patients sent to specialized medical care which may limit the generalisability of the findings. Information on, for example, the mildest depressions that have been treated in primary health care or occupational health care was not necessarily available. For this particular study, we had access to the sum of comorbidities for each patient and data on previous depression diagnoses. No information was available on whether the patient was still suffering from depression. The same limitation applies to other comorbidities. For technical reasons, we were unable to distinguish between temporary diagnoses and long-term and permanent diagnoses. Lastly, we did not account for differences across race or ethnicity. However, the population is known to be very homogenous especially in the two oldest age groups.

## Conclusion

This study revealed that some of the clinical features found in the young-middle aged (< 70 years old) and in the elderly (70–80 years old) are also present in the very elderly (>80 years old). The claim about less daytime sleepiness in the elderly sleep apnea patients does not seem to be true in patients with at least moderate sleep apnea.

While the severity of sleep apnea did not differ between age groups based on any of the cardiorespiratory polygraphy variables studied, there were differences in perceived symptoms.

Occurrence of mental distress was higher in the very elderly, suggesting higher likelihood of psychiatric conditions. Levels of depressive symptoms in women were consistently higher than in men of comparable age including the very elderly, which is in line with the previous studies.

We found a significant gap in the number of depression diagnoses and high scores of the DEPS questionnaire mapping the occurrence of depression which can mean a significant amount of undiagnosed depression, especially in the elderly and very elderly. We showed the link between occurrence of depressive symptoms and age, female gender, CB-$pCO_2$, ESS score and mean $SpO_2$.

Sleep apnea should not be ignored in older adults and studies focusing on specific characteristics and outcome as well as effectiveness of CPAP treatment in older adults are needed.

## Supporting information

**S1 Table. The proportion of GHQ-12 score ≥ 3 among sleep apnea patients by age group and gender.**
(DOCX)

**S2 Table. The proportion of DEPS score ≥ 9 among sleep apnea patients by age group and gender.**
(DOCX)

## Author contributions

**Conceptualization:** Aino Lammintausta, Per-Erik Gustafsson, Ulla Anttalainen, Tarja Saaresranta.

**Data curation:** Olli Lyyra, Aino Lammintausta, Per-Erik Gustafsson, Ulla Anttalainen, Tarja Saaresranta.

**Formal analysis:** Olli Lyyra, Aino Lammintausta, Per-Erik Gustafsson, Ulla Anttalainen, Tarja Saaresranta.

**Funding acquisition:** Olli Lyyra, Aino Lammintausta, Ulla Anttalainen, Tarja Saaresranta.

**Investigation:** Olli Lyyra, Aino Lammintausta, Per-Erik Gustafsson, Ulla Anttalainen, Tarja Saaresranta.

**Methodology:** Olli Lyyra, Aino Lammintausta, Per-Erik Gustafsson, Ulla Anttalainen, Tarja Saaresranta.

**Project administration:** Ulla Anttalainen, Tarja Saaresranta.

**Resources:** Olli Lyyra, Aino Lammintausta, Ulla Anttalainen, Tarja Saaresranta.

**Software:** Olli Lyyra, Aino Lammintausta.

**Supervision:** Olli Lyyra, Aino Lammintausta, Ulla Anttalainen, Tarja Saaresranta.

**Validation:** Olli Lyyra, Aino Lammintausta, Ulla Anttalainen, Tarja Saaresranta.

**Visualization:** Olli Lyyra, Aino Lammintausta, Ulla Anttalainen, Tarja Saaresranta.

**Writing – original draft:** Olli Lyyra, Aino Lammintausta, Per-Erik Gustafsson, Ulla Anttalainen, Tarja Saaresranta.

**Writing – review & editing:** Olli Lyyra, Aino Lammintausta, Ulla Anttalainen, Tarja Saaresranta.

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
