## [Decision Letter · Decision Letter 0]

21 Dec 2024

PONE-D-24-21361Depressive symptoms are more common in elderly sleep apnea patientsPLOS ONE

Dear Dr. Lyyra,

Thank you for submitting your manuscript to PLOS ONE. After careful consideration, we feel that it has merit but does not fully meet PLOS ONE’s publication criteria as it currently stands. Therefore, we invite you to submit a revised version of the manuscript that addresses the points raised during the review process.

We look forward to receiving your revised manuscript.

Kind regards,

Mohammad Hossein Ebrahimi

Academic Editor

PLOS ONE

Journal Requirements:

Reviewers' comments:

Reviewer's Responses to Questions

**Comments to the Author**

1. Is the manuscript technically sound, and do the data support the conclusions?

Reviewer #1: Yes

Reviewer #2: Yes

2. Has the statistical analysis been performed appropriately and rigorously? 

Reviewer #1: Yes

Reviewer #2: Yes

3. Have the authors made all data underlying the findings in their manuscript fully available?

Reviewer #1: Yes

Reviewer #2: Yes

4. Is the manuscript presented in an intelligible fashion and written in standard English?

Reviewer #1: Yes

Reviewer #2: Yes

5. Review Comments to the Author

Reviewer #1: The manuscript "Depressive symptoms are more common in elderly sleep apnea patients" by Olli Lyyra et al. addresses a topic that has been relatively underexplored, namely sleep apnea in the elderly, and highlights how mental well-being can impact the worsening of this clinical condition. This work confirms and expands our understanding of how mood and behavioral disorders can exacerbate and possibly induce sleep apnea. In my opinion, the manuscript is worthy of publication.

Reviewer #2: This study offers important insights into sleep apnea in older adults, with a particular focus on gender differences. By analyzing a large group including a significant number of very elderly individuals, it adds valuable data to existing research. A key finding is that while the severity of sleep apnea didn’t vary much across age groups, symptoms like daytime sleepiness and mental distress did. The study also sheds light on the underdiagnosis of depression, particularly women. By exploring factors like SpO2 and ESS scores, it deepens our understanding of the connections between sleep apnea and over mental health.

Major Issues:

The title "Depressive symptoms are more common in elderly sleep apnea patients" does not accurately reflect the full scope of the study.

Please provide an explanation and, if feasible, the rationale behind dividing your population in this manner, along with the sources that support this approach. For example, the United Nations defines an older person as someone over 60 years of age, while the most commonly used age to define elderly is 65.

Provide a more detailed theoretical explanation for the gap in depression findings in your study, as this represents a significant limitation.

Minor Issues:

The conclusion section needs better alignment with the organization of the results, particularly regarding the occurrence of depressive symptoms in women (lines 47), which should be explicitly addressed in the results section.

The keywords overlook important terms like depression and psychological distress, which are essential for a comprehensive understanding of the topic.

A more critical examination is needed to assess the impact of using polysomnography, given its application in less than 1% of patients.

Offer a more detailed explanation of the distinction between clinically diagnosed depression and self-reported symptoms, highlighting the differences in diagnostic processes, criteria, and their implications for treatment.

Please incorporate age subtypes—very elderly, elderly, young, and middle-aged—into each table, and Replace 'Depression' with 'DEPS' in Table 5.

Add more information to contextualize your research and provide a clearer direction for future research, including potential areas for further exploration and development.

While the conclusion is clear and concise, it is somewhat lengthy

6. PLOS authors have the option to publish the peer review history of their article (what does this mean? ). If published, this will include your full peer review and any attached files.

**Do you want your identity to be public for this peer review?** For information about this choice, including consent withdrawal, please see our Privacy Policy .

Reviewer #1: **Yes: ** Salvatore Rinaldi

Reviewer #2: No

---

## [Author Response · Author response to Decision Letter 1]

13 Jan 2025

We would kindly like to thank the reviewers for devoting their valuable time to review our manuscript. Our detailed responses to the specific comments can be found below in italics.

Reviewer #1: The manuscript "Depressive symptoms are more common in elderly sleep apnea patients" by Olli Lyyra et al. addresses a topic that has been relatively underexplored, namely sleep apnea in the elderly, and highlights how mental well-being can impact the worsening of this clinical condition. This work confirms and expands our understanding of how mood and behavioral disorders can exacerbate and possibly induce sleep apnea. In my opinion, the manuscript is worthy of publication.

We would like to thank the reviewer for these positive and encouraging comments.

Reviewer #2: This study offers important insights into sleep apnea in older adults, with a particular focus on gender differences. By analyzing a large group including a significant number of very elderly individuals, it adds valuable data to existing research. A key finding is that while the severity of sleep apnea didn’t vary much across age groups, symptoms like daytime sleepiness and mental distress did. The study also sheds light on the underdiagnosis of depression, particularly women. By exploring factors like SpO2 and ESS scores, it deepens our understanding of the connections between sleep apnea and over mental health.

We would sincerely like to thank the reviewer for these kind words and the constructive comments regarding the manuscript.

Major Issues:

The title "Depressive symptoms are more common in elderly sleep apnea patients" does not accurately reflect the full scope of the study.

We thank the reviewer for pointing out this issue. We have now modified the title as follows: “Differences in the clinical presentation of sleep apnea patients according to age and gender”

Please provide an explanation and, if feasible, the rationale behind dividing your population in this manner, along with the sources that support this approach. For example, the United Nations defines an older person as someone over 60 years of age, while the most commonly used age to define elderly is 65.

Finnish people frequently work after their retirement age since it increases their pension. The old-age pension will increase by 0.4 per cent for each month that a employee retires late, that is, for each month past the month in which he/she reached his/her retirement age of 69 when the employer’s statutory pension insurance obligation ends. Therefore people 60 to 65 and even 65-70 years old are often still very active both physically and socially. The proportion of frail elderly people rises more thereafter. Based on these points, we decided to use the age of 70 as a cut-off point.

Provide a more detailed theoretical explanation for the gap in depression findings in your study, as this represents a significant limitation.

We have extended the Discussion, beginning from page 14, line 329 as follows: “It is possible that a larger proportion depression remains undiagnosed in the eldelry, perhaps due to prejudice that depression is a normal part of ageing. Indeed, a large Brazilian National Survey with 70,806 participants (15-107 years old) found that the prevalence of depression underdiagnosis was higher in the elderly population.” Faisal-Cury A, Ziebold C, Rodrigues DMO, Matijasevich A. Depression underdiagnosis: Prevalence and associated factors. A population-based study. J Psychiatr Res. 2022 Jul;151:157-165. doi: 10.1016/j.jpsychires.2022.04.025.

Minor Issues:

The conclusion section needs better alignment with the organization of the results, particularly regarding the occurrence of depressive symptoms in women (lines 47), which should be explicitly addressed in the results section.

We have revised the Abstract’s Results -section to better represent the differences in the elderly population as follows: “The severity of sleep apnea did not differ between the age-groups based on any of the PG variables studied. No significant differences wer found in the level of subjective daytime sleepiness between age-groups. Women had higher DEPS scores than men in all age-groups. Very elderly men had higher DEPS scores compared to men in other age-groups (6.3 ±4.6 vs. 5.6 ±5.9 vs. 5.1 ±4.8, p<0.05) while the differences in DEPS scores did not reach significance among women. Each unit increase in SpO2 was associated with a 22% decrease in the odds of having a DEPS score of ≥9.”

Further, we have revised the Conclusion as follows: “The severity of sleep apnea or subjective daytime sleepiness did not differ among age-groups in moderate-severe sleep apnea patients. Occurrence of depressive symptoms was consistently more common in women than in men of comparable age. Mental wellbeing was the worst in the very elderly. Higher SpO2 was associated with less depressive symptoms.”

The keywords overlook important terms like depression and psychological distress, which are essential for a comprehensive understanding of the topic.

Thank you for this comment. We have added the terms “daytime sleepiness”, “depression” and psychological distress“ to the list of keywords.

A more critical examination is needed to assess the impact of using polysomnography, given its application in less than 1% of patients.

We have added the following text in limitations of the study, beginning from page16, line 374: “PSG is the gold standard when diagnosing sleep apnea, and more accurately determines the severity of sleep apnea. Therefore, we cannot exclude that using PG may overlook or weaken some differences between the age groups or genders.”

Offer a more detailed explanation of the distinction between clinically diagnosed depression and self-reported symptoms, highlighting the differences in diagnostic processes, criteria, and their implications for treatment.

We have now provided a more detailed explanation as follows, beginning from page 14, line 310: “However, the actual diagnosis of depression cannot be based only on symptom questionnaires. Self-reported symptoms do not reliably differentiate between transient depressive symptoms and DSM-5 criteria of depression. The diagnosis of depression is always based on the symptoms found during the interview, the number, severity, duration and temporal predominance of them, and ruling out differential diagnoses with physical examination and laboratory tests.”

Please incorporate age subtypes—very elderly, elderly, young, and middle-aged—into each table, and Replace 'Depression' with 'DEPS' in Table 5.

We have incorporated age subtypes into each table, and replaced “Depression” with “DEPS” in Table 5.

Add more information to contextualize your research and provide a clearer direction for future research, including potential areas for further exploration and development.

We have added the following text in Discussion on page 16: “Our study highlights the impact of age and gender on the presentation of sleep apnea in the elderly population. More detailed exploration of different presentation of phenotypes and their impact on treatment adherence as well as treatment efficacy is needed in order to focus the limited healthcare resources cost-effectively.”

While the conclusion is clear and concise, it is somewhat lengthy.

We have shortened the conclusions as follows:” This study revealed that some of the clinical features found in the young-middle aged (< 70 years old) and in the elderly (70 -80 years old) are also present in the very elderly (>80 years old). The claim about less daytime sleepiness in the elderly sleep apnea patients does not seem to be true in patients with at least moderate sleep apnea.

While the severity of sleep apnea did not differ between age groups based on any of the cardiorespiratory polygraphy variables studied, there were differences in perceived symptoms.

Occurrence of mental distress was higher in the very elderly suggesting higher likelihood of psychiatric conditions. Levels of depressive symptoms in women were consistently higher than in men of comparable age including the very elderly, which is in line with the previous studies.

We found a significant gap in the number of depression diagnoses and high scores of the DEPS questionnaire mapping the occurrence of depression which can mean a significant amount of undiagnosed depression, especially in the elderly and very elderly. We showed the link between occurrence of depressive symptoms and age, female gender, CB-pCO2, ESS score and mean SpO2.

Sleep apnea should not be ignored in older adults and studies focusing on specific characteristics and outcome as well as effectiveness of CPAP treatment in older adults are needed.”

---

## [Editor Report · Decision Letter 1]

19 Jan 2025

Differences in the clinical presentation of sleep apnea patients according to age and gender

PONE-D-24-21361R1

Dear Dr. Lyyra,

We’re pleased to inform you that your manuscript has been judged scientifically suitable for publication and will be formally accepted for publication once it meets all outstanding technical requirements.

Kind regards,

Mohammad Hossein Ebrahimi

Academic Editor

PLOS ONE
---

## [Editor Report · Acceptance letter]

PONE-D-24-21361R1

PLOS ONE

Dear Dr. Lyyra,

I'm pleased to inform you that your manuscript has been deemed suitable for publication in PLOS ONE. Congratulations! Your manuscript is now being handed over to our production team.

Kind regards,

on behalf of

Dr. Mohammad Hossein Ebrahimi

Academic Editor

PLOS ONE